# Risk of Hepatocellular Carcinoma in Patients with Porphyria: A Systematic Review

**DOI:** 10.3390/cancers14122947

**Published:** 2022-06-15

**Authors:** Daryl Ramai, Smit S. Deliwala, Saurabh Chandan, Janice Lester, Jameel Singh, Jayanta Samanta, Sara di Nunzio, Fabio Perversi, Francesca Cappellini, Aashni Shah, Michele Ghidini, Rodolfo Sacco, Antonio Facciorusso, Luca Giacomelli

**Affiliations:** 1Division of Gastroenterology and Hepatology, University of Utah, Salt Lake City, UT 84112, USA; daryl.ramai@hsc.utah.edu; 2Department of Internal Medicine, Hurley Medical Center, Michigan State University, Flint, MI 48503, USA; deliwal1@msu.edu; 3Division of Gastroenterology & Hepatology, CHI Health Creighton University Medical Center, Omaha, NE 68131, USA; saurabhchandan@gmail.com; 4Health Science Library, Long Island Jewish Medical Center, Northwell Health, New Hyde Park, NY 11040, USA; jlester1@northwell.edu; 5Department of Internal Medicine, Mather Hospital, Northwell Health, Port Jefferson, NY 11777, USA; jameel.k.singh@gmail.com; 6Department of Gastroenterology, Postgraduate Institute of Medical Education and Research, Chandigarh, Sector 12, Chandigarh 160012, India; dj_samanta@yahoo.co.in; 7Polistudium SRL, 20135 Milan, Italy; sara.dinunzio@polistudium.it (S.d.N.); fabio.perversi@polistudium.it (F.P.); francesca.cappellini@polistudium.it (F.C.); aashni.shah@polistudium.it (A.S.); 8Division of Medical Oncology, Fondazione IRCCS Ca’ Granda, Ospedale Maggiore Policlinico, 20122 Milan, Italy; michele.ghidini@policlinico.mi.it; 9Gastroenterology Unit, Department of Surgical and Medical Sciences, University of Foggia, 71122 Foggia, Italy; r.sacco@ao-pisa.toscana.it (R.S.); antonio.facciorusso@virgilio.it (A.F.)

**Keywords:** porphyria cutanea tarda, porphyria, ADA dehydratase deficiency porphyria, acute intermittent porphyria, hepatocellular carcinoma, cirrhosis

## Abstract

**Simple Summary:**

Porphyria is a metabolic condition which leads to reduced heme production. While it involves multiple organs systems, porphyria affecting the liver can lead to elevations in hepatic enzymes, progressive fibrosis, cirrhosis and eventually malignancy. Our study looked at the prevalence of liver cancer in patients with porphyria. Overall, we found that patients with porphyria are at increased risk of developing hepatic malignancy. As a result, patients with porphyria should undergo routine surveillance for detecting primary liver malignancy.

**Abstract:**

Acute porphyrias are a group of metabolic disorders resulting in defective porphyrin synthesis and reduced heme production, which carries a risk of malignancy. Porphyrias are inborn defects in the heme biosynthesis pathway resulting in neurovisceral manifestations and cutaneous photosensitivity attacks with multi-systemic involvement. Its estimated prevalence nears 5 per 100,000 patients worldwide. Subclinical liver disease is common, which can progress into transaminitis, fibrosis, cirrhosis, and malignancy. However, data on the incidence of primary liver cancer are lacking. We aim to determine the risk of hepatocellular carcinoma (HCC) in patients with porphyria. A systematic review and pooled analysis were conducted through 2021 on studies assessing blood tests, imaging, cancer development, liver transplant, surgical resection, and outcomes in porphyria. In total, 19 studies, which included 7381 patients with porphyria (3476 females), were considered for the final review. In eight studies, alpha-fetoprotein levels were elevated between 200 and 1000 IU/mL. Of the total cohort of patients with porphyria, primary liver cancer was diagnosed in 351 patients (4.8%), of whom 243 (3.3% of the total) were found to have HCC. A subset of patients was found to have cholangiocarcinoma (*n* = 18; 0.3% of the total). Interestingly, advanced liver disease or cirrhosis was not a prerequisite for the formation of HCC in a small group of patients. Of the total cohort, 30 patients underwent liver resection, 48 patients underwent liver transplantation, and 327 patients died. Patients with porphyria are at risk of developing primary liver malignancy. Further studies should aim to develop diagnostic and prognostic models aimed at the early detection of HCC in porphyria.

## 1. Introduction

Amongst liver cancers, hepatocellular carcinoma (HCC) is the most common—accounting for over 90% of all cases [1,2]. Over the years, multiple risk factors for the development of liver cancer have been identified; however, chronic liver disease and the setting of cirrhosis remain the most potent factors.

Porphyrias are inborn errors in the heme biosynthesis pathway resulting in neurovisceral manifestations and cutaneous photosensitivity attacks with multi-systemic involvement. Its estimated prevalence nears 5 per 100,000 and is classified into hepatic or erythropoietic based on whether the pathway intermediates accumulate in the liver or the bone marrow [2,3]. Acute hepatic porphyrias (AHP) are a group of four ultrarare metabolic disorders: 5-aminolevulinic acid (ALA) dehydratase deficiency porphyria (ADP), acute intermittent porphyria (AIP), hereditary coproporphyria (HCP), and variegate porphyria (VP), resulting in defective porphyrin synthesis and reduced heme production [3]. Clinically, porphyrias manifest with chronic gastrointestinal, neurovisceral, and cutaneous manifestations presenting as life-threatening attacks and negatively impact daily functioning and quality of life. Subclinical liver disease is common and manifests as progressive transaminitis, fibrosis, cirrhosis, or HCC leading to premature death. Guidance statements have recommended HCC screening beginning at the age of 50 years [4].

A recent European study reported an annual incidence of HCC 0.07%, with a 95% CI upper limit of 0.17% [5]. According to the European Association for the Study of the Liver (EASL) and the European Organization for Research and Treatment of Cancer (EORTC), clinical practice guidelines for the management of HCC, an annual incidence of 1.5% would warrant surveillance of HCC in cirrhotic patients and 0.2% would warrant surveillance in non-cirrhotic patients, supporting a rationale for an active surveillance program in this cohort [5]. At the same time, the American Association for the Study of Liver Diseases (AASLD) guidelines recommend liver ultrasound with or without alfa-fetoprotein (AFP) evaluation every 6 months for HCC surveillance in patients with cirrhosis without porphyria. Due to the rarity of porphyrias, the role of AFP and HCC surveillance in AHP remains poorly explored [6].

Although HCC incidence varies greatly worldwide, it affects population cohorts with specific predisposing environmental, heritable, and acquired factors. Most of the experience of primary liver cancers amongst patients with porphyria comes from Scandinavian studies, with cirrhosis in up to 40% of the highest risk cohorts. Recent studies have noted an increasing incidence of HCC and porphyria amongst USA cohorts. At the same time, our current understanding of these two entities comes from small observational studies [4,7,8,9,10,11,12].

With this in mind, we aim to perform a systematic review to understand the pooled burden of HCC in patients with porphyria and guide high-risk groups.

## 2. Methods

### 2.1. Search Strategy

We conducted a comprehensive search of several databases, including PubMed, EMBASE, and Web of Science, last updated in February 2022. An experienced medical librarian using inputs from the studies’ authors helped with the literature search. We followed the Preferred Reporting Items for Systematic Reviews and Meta-Analyses (PRISMA) guidelines (Appendix A).

The keywords used in the literature search included but were not limited to a combination of “porphyria,” “hepatic porphyria,” “delta-aminolevulinate-dehydratase deficiency porphyria,” “acute intermittent porphyria,” “porphyria cutanea tarda,” “cirrhosis,” “hepatocellular carcinoma,” “liver cancer,” and “liver transplant,” to identify relevant articles in English. Further details of the search strategy can be found in Appendix A. The search was restricted to studies performed on human subjects and published in English in peer-reviewed journals.

### 2.2. Study Selection

We included studies reporting on the risk and rate of HCC in patients with a history of porphyria. Studies were included irrespective of inpatient/outpatient setting and geographical setting if they reported the clinical outcomes and data needed for analysis.

The inclusion criteria were as follows: (1) patients ≥ 18 years with a medical history of porphyria; and (2) studies reporting the risk or development of hepatocellular carcinoma. The exclusion criteria comprised: (1) pediatric (age < 18 years) studies; and (2) case reports or case series with <10 patients. In the event of multiple publications from the same cohort or overlapping cohorts, data from the most recent and/or most appropriate comprehensive report were retained.

### 2.3. Data Abstraction

Study references and citations were collected in EndNote X9 (Thomson Reuters, New York, NY, USA). The Covidence systematic review software (Veritas Health Innovation, Melbourne, Australia) was used to further screen and extract relevant studies. The full text of each selected article was reviewed to verify that it contained relevant information. The bibliographic section of the selected articles was manually searched for additional relevant articles to identify other potentially eligible publications. Data on study-related outcomes in the individual studies were abstracted by two authors (Daryl Ramai, Smith Deliwala). The Newcastle–Ottawa scale was used for cohort studies.

### 2.4. Quality Assessment

Two reviewers (Smith Deliwala and Saurabh Chandan) independently carried out quality scoring. Non-randomized studies were assessed via Newcastle–Ottawa scale (NOS). No additional assessment tools were necessary as no randomized controlled studies were based on our literature search. Each study was assessed independently across three areas of potential bias: patient selection, comparability, and outcome reporting. The score range of NOS is from 0 to 9. A score ≥ 7 was considered high quality, 4–6, high risk, and 0–3, very high risk of bias.

## 3. Results

### 3.1. Search Results

From an initial search showing 840 studies, 282 duplicates were removed, and 558 were screened. Two studies were excluded due to having multiple publications from the same cohort, of which only the most recent publication was included [9,13]. Ultimately, 19 studies were included in the final systematic review and analysis (Figure 1).

### 3.2. Study Characteristics

In total, 7381 patients with porphyria, of whom 3476 were female, were included in the final review. However, details on the subtype of porphyria were not consistently reported across studies. The patients had a broad age range (30–80 years) at diagnosis. Ten studies were retrospective cohorts; three were case–control studies, five were prospective cohorts, and one study was a cross-sectional questionnaire. The median follow-up time ranged from 3 to 24 years. Nine studies were conducted in Scandinavia (Sweden—6, Norway—1, Finland—1, and Denmark—1), three studies were conducted in Spain, two studies were conducted in Italy, and one each from The Netherlands, Belgium, Germany, France, and the USA. Additional study and patient characteristics are detailed in Table 1.

### 3.3. Characteristics of Patients with HCC

Of the total cohort of patients with porphyria, primary liver cancer was diagnosed in 351/7381 patients (4.8%), of whom 243 patients (3.3%) were found to have HCC. At the time of HCC diagnosis, most patients were over 50 years of age. A small subset of patients was found to have cholangiocarcinoma (*n* = 18 or 0.3%).

Interestingly, we found that not all cases of HCC progression were preceded by advanced liver fibrosis or cirrhosis. Andant et al., reported seven patients who had HCC but no evidence of fibrosis [4]. Sardh et al., described 20 patients with HCC and three patients with cholangiocarcinoma, of whom three had stage I fibrosis, six had stage II fibrosis, and four had stage IV fibrosis [10]. The fibrosis stage was not reported for the remaining cases. Saberi et al., reported five patients with HCC; however, only one patient was noted to have stage II fibrosis [6].

### 3.4. Liver Resection

Liver resection was the most common treatment of porphyria-associated HCC. Innala et al., reported that in porphyria patients who were regularly screened for HCC (*n* = 8) versus those who were not (*n* = 14), seven patients (87.5%) of screened patients underwent liver resection with a recurrence rate of 42.9% (*n* = 3) [8]. Compared to those who did not undergo periodic screening, only four patients (28.6%) underwent tumor resection with a recurrence rate of 75% (*n* = 3/4) (Table 2). It should be noted that tumor burden (>7 cm) was higher in the unscreened group (*n* = 10).

Andant et al., reported that six patients (from a cohort of 650) underwent resection, of which 50% (*n* = 3) of the patients had cancer recurrence with an average recurrence time of 1.5 years and an average follow-up of 7 years [4]. Similarly, Fracanzani et al., reported that 7 out of 53 patients underwent resection; however, no recurrence was reported at an average follow-up time of 6 ± 4.5 years [20]. Saberi et al., reported that among 327 patients with AHP, 5 (1.5%) were diagnosed with HCC, of whom 4 patients underwent resection (three with resection only; one with both resection and locoregional therapy). However, 50% (*n* = 2) of the patients developed HCC recurrence [6]. Sardh et al., reported that in a cohort of 179 patients, 23 patients were diagnosed with primary liver cancer (12 patients were detected by annual radiology surveillance). Of these, nine patients underwent resection, and two patients from this cohort had the longest survival times (12.5 and 13.3 years) among all patients [10].

### 3.5. Liver Transplant

Only a minority of patients underwent a liver transplant. Elder et al., reported that from their study cohort of 335 patients, 56 patients (16.7%) underwent a liver transplant. Of these patients, no cancer recurrence was reported at a mean follow-up time of 3 years [7].

### 3.6. Quality Assessment

Two studies were rated as high quality, while all other studies were regarded as having a high risk of bias according to the NOS scale. Studies were downrated due to missing information such as follow-up time, treatment outcomes, recurrence, and mortality. A detailed assessment of study quality is given in Appendix A.

## 4. Discussion

The results of our systematic review suggest that patients with AHP had an estimated 5% risk of primary liver cancer compared with the average population, where the disease burden ranges from <3.3 to >8.4 per 100,000 persons globally based on substantial variations in infectious and environmental risk factors [25,26]. Interestingly, our review showed that HCC could occur even in the absence of cirrhosis, unlike other causes of chronic liver disease. Our findings highlight the risk of HCC in this patient cohort and support current guidelines of liver imaging at 6–12 month intervals after the age of 50 years [27]. The pathogenesis of HCC development in AHP is not well understood but is thought to bypass the fibrosis–cirrhosis–cancer progression pathway.

Indeed, ALA and porphobilinogen (PBG) accumulation has toxic effects on hepatocytes, leading to free radical formation, hepatocyte injury, and subsequent DNA damage [27]. Elevated urinary ALA, PBG, and other uroporphyrins were seen in the identified studies, supporting this hypothesis [6,14,16,17,18,22,23,24,25,27,28,29,30].

AFP levels were found to be elevated. In our review, studies reported AFP levels between 200 and 1000 IU/mL, much higher than in other chronic liver diseases, suggesting that there may be a role for this marker in HCC screening [4,6,8,14,17,20]. Despite primary liver cancer being the most reported cancer in patients with AHP, recent studies have reported an association with cholangiocarcinoma, suggesting extra-hepatic manifestations of this disease [10,15].

The management of AHP begins with identifying and eliminating the factor that leads to attacks, including alcohol use, prolonged fasting, crash dieting, smoking, or medication use. Patients should inform physicians about their diagnosis during pre-operative evaluations, especially in patients with frequent attacks that may require prophylactic hemin infusions. All patients that are fasting should be given dextrose infusions routinely.

The prevention of recurrent episodes can be achieved using prophylactic or “on-demand” infusions of hemin [28]. Carbohydrate loading with glucose tablets or dextrose solutions during the early stages of the attack has not shown any clear benefit [26]. A small cohort of patients with AHP that develop cyclic attacks related to their menstrual cycle has benefited from early gonadotropin-releasing hormone analogs [30]. Despite hemin therapy, orthotopic liver transplantation has been successful in severe, refractory cases. However, patients with advanced neuropathy or respiratory paralysis are considered poor candidates [31]. Concomitantly, our review showed that liver resection was used in a significant number of patients. To this end, a small number of patients experienced a recurrence of HCC, which emphasizes the importance of continued surveillance in this population.

This study provides further evidence to the current understanding of HCC in patients with AHP. This extensive review includes 19 studies from multiple countries, supporting screening guidelines in this patient population. Limitations include confounding lifestyle risk factors such as obesity or alcohol use were not reported in many studies. While there is a favorable number of included studies, a significant proportion were retrospective cohorts, hence prone to bias. Further, some studies had incomplete or missing data, which led to significant heterogeneity; hence, meta-analytic methods were not employed. Furthermore, our systematic review only included English articles; thus, the generalization of our findings to non-English countries and patients may be limited. Despite these limitations, this is the first comprehensive review to assess the risk of HCC in patients with porphyria.

## 5. Conclusions

Our systematic review suggests that patients with porphyria hold a 5% risk for the development of primary liver cancer, respectively, higher than previous estimates. HCC was found in 3.3% of the study cohort. AFP should be used in conjunction with imaging modalities to screen for HCC after the age of 50 years as frequently as every 6–12 months. Further research should address the development of clinical models for the early detection of HCC in patients with different phenotypes of porphyria.

## Figures and Tables

**Figure 1 cancers-14-02947-f001:**
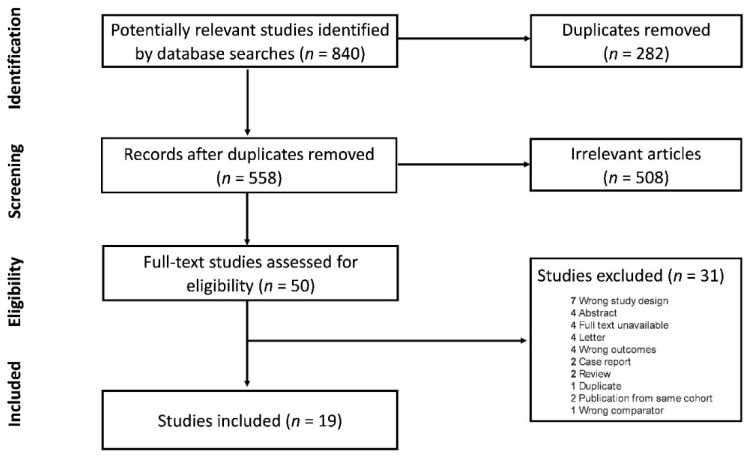
Flow chart of included studies.

**Table 1 cancers-14-02947-t001:** Characteristics of the porphyria study population.

Author/Year	Design	Location	Total Patients (*n*)	Total female *n* (%)	Age (Years), Mean ± SD	Age at HCC Diagnosis (Years), Mean ± SD	Urinary Cutoffs	α-Fetoprotein Levels	Cancer Type (*n*)	Porphyria Subtype with Liver Cancer (%)
Solis 1982 [14]	Prospective, single-center	Spain	138	3 (2)	63.5 ± 6.6	NR	URO (742 ± 437 µg/day) COPRO (310 ± 147 µg/day)	780 ng/mL 1320 ng/mL 2150 ng/mL Positive (5) ND (2)	HCC (7) Unknown (3)	PCT 10 (7.2)
Hardell 1984 [15]	Case–control, single-center	Sweden	103	0	Range 30–80	67 ± 1.7	NR	NR	HCC (83) ICC (15) HSA (3) Unspecified sarcoma (1)	AIP 3 (3.6) PCT 6 (7.2)
Lithner 1984 [16]	Retrospective, single-center	Sweden	206	120 (58)	55	>65	Elevated levels of PBG (137)	NR	HCC (11)	AIP 11 (5.3%)
Salata 1985 [17]	Prospective, single-center	Spain	83	6 (7.2)	57.4	59.5 ± 5.3	COPRO < 204 URO < 18 nmol/l	Elevated in 3 out of 9 HCC cases	HCC (13)	PCT 13 (15.7)
Siersema 1992 [18]	Prospective, single-center	The Netherlands	38	13 (34)	48 ± 12	53.8 ± 4.3	URO < 4.0 nmol/mmol creatinine HCP< 3.2 nmol/mmol creatinine	None were elevated	HCC (5)	PCT 5 (15)
Kauppinen 1992 [11]	Retrospective, single-center	Finland	206	121 (58.7)	49 ± 19	NR	NR	NR	HCC (8)	AIP 7 (88) VP 1/8 (12)
Andersson 1996 [12]	Retrospective, single-center	Sweden	2122	852 (40)	27 ± 10	71 ± 12	NR	NR	HCC (9)	AIP 9 (27)
Linet 1999 [19]	Retrospective, multicenter	Denmark	826	392 (48)	62 ± 18	62 ± 6.6	NR	NR	HCC (12)	PCT 7 (58) AIP 5 (42)
Andant 2000 [4]	Prospective, single-center	Italy	650	347 (53)	41 ± 7	50 ± 9	ALA: 7.2 ± 1.5 PBG: 4 ± 1.3	>200 IU/mL (7)	HCC (7)	AIP 5 (71) VP 1 (14) HC 1 (14)
Fracanzani 2001 [20]	Case–control, single-center	Italy	53	2 (3.8)	56 ± 8	NR	URO (3607 ± 1850 µg/24 h)	>400 UI/mL (1)	HCC (18)	PCT 18 (51)
Gisbert 2004 [21]	Prospective, Single-center	Spain	39	4 (10)	55 ± 16	69	NR	Elevated (1)	HCC (1)	PCT 1 (2.6)
Cassiman 2008 [22]	Retrospective, single-center	Belgium	17	7 (41)	43 ± 3	NR	NR	NR	HCC (1)	PCT 1 (5.9)
Innala 2011 [8]	Case–control, single-center	Sweden	81	49 (60.5)	67 ± 9.5	70 ± 6.5	ALA <45 µmol/L) PBG <11 µmol/L	>20–199 (1) 200–1000 (3) >1000 (1)	HCC (22)	AIP 22 (27)
Sardh 2013 [10]	Retrospective, single-center	Sweden	179	111 (62)	>50	66.1 ± 8.6	PBG < 1.2 mmol/mol creatinine ALA < 3.1 mmol/mol creatinine	Elevated but < 200 ng/mL (2)	HCC (20) CC (3)	AIP 17 (85) VP 2 (10) HC 1 (5) AIP 3 (100)
Elder 2013 [7]	Retrospective multicenter	France	335	213 (63.6)	37 ± 20	NR	PBG (UK): 13 (10–213) µmol/mmol creatinine PBG (France): 21 (13–44) µmol/mmol creatinine	NR	HCC (14)	AIP 11 (78.6) VP 3 (21)
Lang 2015 [23]	Questionnaire	Germany	122	NR	NR	NR	NR	NR	HCC (1)	AIP 1 (0.82)
Baravelli 2019 [5]	Retrospective multicenter	Norway	612	319 (52)	52 ± 13	NR	URO: 30 nmol/mmol creatinine	NR	HCC (6)	PCT 5 (0.85)
Saberi 2020 [6]	Retrospective, multicenter	USA	327	266 (81)	32 ± 5	69 ± 5	PBG: 8 mg/24 h or >2× creatinine	<10 ng/mL (4)	HCC (5)	AIP 4 (80) VP 1 (20)
Lissing 2022 [24]	Retrospective	Sweden	1244	654 (53)	Median (range) 36 (19–53)	Median (range) 71 (53–89)	PBG > 1.6 mmol/mol creatinine	NR	PLC (83): HCC (67), CC (3), and unspecified (13)	AIP 81 (7.6) VP 1 (0.8) HCP 1 (1.8)

AIP: Acute intermittent porphyria; VP: variegate porphyria; PCT: porphyria cutanea tarda; COPRO: Coproporphyrin; HCC: HCP: Hereditary coproporphyria; PBG: Prophobilinogen; URO: uroporphyrin. HCP: heptacarboxylporphyrins; Hepatocellular carcinoma; ICC: Intrahepatic Cholangiocarcinoma, CC: Cholangiocarcinoma, HSA: Hemangiosarcoma, NR: Not reported, ND: Not Done.

**Table 2 cancers-14-02947-t002:** Details of treatment and outcomes in patients with porphyria.

Author/Year	Liver Transplant	Other Treatments (n)	Recurrence (Yes/No)	Time to Recurrence (Years), Mean ± SD	Follow-Up Time (Years), Mean ± SD	Death (*n*)
Solis 1982 [14]	NR	No	NR	NR	4.7	9
Hardell 1984 [15]	NR	No	NR	NR	4806 occupation years	NR
Lithner 1984 [16]	NR	No	NR	NR	20	11 (HCC patients), total number of deaths are unknown
Salata 1985 [17]	NR	No	NR	NR	4.8 ± 3.5	NR
Siersema 1992 [18]	NR	No	NR	NR	9.9 ± 5.4	NR
Kauppinen 1992 [11]	NR	No	NR	NR	13.7 ± 7.4	96
Andersson 1996 [12]	NR	No	NR	NR	12	33
Linet 1999 [19]	NR	No	NR	NR	3	10
Andant 2000 [4]	No	Resection (6)	Yes (3)	1.7 ± 1	7	4
Fracanzani 2001 [20]	No	Resection (7)	No	NR	6 ± 4.5	0
Gisbert 2004 [21]	No	NR	NR	NR	9.7 ± 9	NR
Cassiman 2008 [22]	No	Phlebotomy (13) chloroquine (3) IFN (1) Ribavirin (1) Nivaquin (1)	No	NR	24	NR
Innala 2011 [8]	No	Resection (10) Resection + RFA (1) RFA (1) Cytostatic (1)	Yes (6)	4.7 ± 4.5	15	14
Sardh 2013 [10]	No	Resection (5) RT (6) Chemotherapy (5) RFA (2) TACE (2) PEIT (1)	Yes (4)	5.4 ± 2.9	4.6	NR
Elder 2013 [7]	56	No	No	NR	3	NR
Lang 2015 [23]	NR	No	NR	NR	NR	NR
Baravelli 2019 [5]	NR	No	NR	NR	16	150
Saberi 2020 [6]	No	Lenvatinib (1) Resection (4) Nivolumab (1) RFA (1) TACE (1)	Yes (2)	NR	7 ± 5	0
Lissing 2022 [24]	NR	NR	NR	NR	19.5	NR

Hepatocellular carcinoma; NR: Not reported; RFA: Radiofrequency ablation; RT: Radiotherapy; IFN: Interferon; TACE: Transarterial chemoembolization; PEIT: Percutaneous ethanol injection therapy.

## Data Availability

Not applicable.

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
