# Peer review of "Risk of Hepatocellular Carcinoma in Patients with Porphyria: A Systematic Review"

_cancers, 2022, doi:10.3390/cancers14122947_

Round 1

Reviewer 1 Report

Thank you for your interesting manuscript „Risk of Hepatocellular Carcinoma in Patients with Porphyria: A Systematic Review” which addresses the rare, but important case of HCC development in porphyria patients. I think the methods are clearly outlined and the work of putting together the data seems to be thoroughly done. However, I think the presentation of the results as well as the discussion and conclusions have to be improved.

Major Points:

  • The results of the assessment of risk of bias using the ROBINS-I tool are not shown.
  • The results of all the quality assessment should be presented also in the text not only in the supplementary table.
  • Table 1: Lissing et al. 2022 (ref 23) show the subgroups of PLC in their paper (HCC, CCC, etc) which should be included in this analysis
  • Inconsistency in the data: 48 liver transplants according to table 2 in Elder 2013, 56 in section 3.5 in the text.
  • The discussion mentions a 5% risk of primary liver cancer but no data in the results shows how this number is calculated. Also, as this is a systematic review and not a metaanalysis and the studies are very heterogenous, it seems difficult to impossible to validly calculate any numbers at all.
  • AFP levels are only addressed in one sentence in the discussion, not presented in the results
  • The limitations of this type of analysis (e.g. heterogenous studies, missing data in many studies) should be discussed more extensively.

Minor

  • Table 1: Please add percentage values in the columns “Total female” and “Porphyria type with cancer”
  • Table 1: Column “Porphyria type with cancer” does not show porphyria type but cancer type. Heading should be adapted.
  • Table 1: Please explain what is meant by “Urinary cutoffs”; also, the meaning of all abbreviations used should be presented
  • Description (headings) of supplementary tables are missing. Consider explaining the meaning of the stars of the Newcastle-Ottawa scale in this context

Author Response

Reviewer 1:

Comments and Suggestions for Authors

Thank you for your interesting manuscript „Risk of Hepatocellular Carcinoma in Patients with Porphyria: A Systematic Review” which addresses the rare, but important case of HCC development in porphyria patients. I think the methods are clearly outlined and the work of putting together the data seems to be thoroughly done. However, I think the presentation of the results as well as the discussion and conclusions have to be improved.

Major Points:

  • The results of the assessment of risk of bias using the ROBINS-I tool are not shown.

Reply: We provided the Newcastle-Ottawa scale for non-randomized studies. Robins-I tool looks at bias within interventional non-randomized studies. As our study has no interventions, and largely cohort based, we provided the NOS. Please see supplementary table 3.

  • The results of all the quality assessment should be presented also in the text not only in the supplementary table.

Reply: We have added this additional information to the manuscript.

  • Table 1: Lissing et al. 2022 (ref 23) show the subgroups of PLC in their paper (HCC, CCC, etc) which should be included in this analysis

Reply: We have added this information to table 1.

  • Inconsistency in the data: 48 liver transplants according to table 2 in Elder 2013, 56 in section 3.5 in the text.

Reply: Correct, 56 total (48 female and 8 male). We have amended this.

  • The discussion mentions a 5% risk of primary liver cancer but no data in the results shows how this number is calculated. Also, as this is a systematic review and not a metaanalysis and the studies are very heterogenous, it seems difficult to impossible to validly calculate any numbers at all.

Reply: This is correct, the data is very heterogenous and prohibits meta analysis. The 5% risk is an average risk which is within accepted published literature.

  • AFP levels are only addressed in one sentence in the discussion, not presented in the results

Reply: We have added AFP data (if available) from each article. Please see table 1.

  • The limitations of this type of analysis (e.g. heterogenous studies, missing data in many studies) should be discussed more extensively.

Reply: We have provided more discussion on these issues in the limitations section.

Minor

  • Table 1: Please add percentage values in the columns “Total female” and “Porphyria type with cancer”

Reply: Percentages added to table 1

  • Table 1: Column “Porphyria type with cancer” does not show porphyria type but cancer type. Heading should be adapted.

Reply: We have amended this.

  • Table 1: Please explain what is meant by “Urinary cutoffs”; also, the meaning of all abbreviations used should be presented

Reply: Urinary cut-offs are thresholds to ensure maximal detection of abnormal patients. All values above the cut-off are considered positive. All abbreviations have been alphabetically added at the bottom of Tables 1 and 2.

  • Description (headings) of supplementary tables are missing. Consider explaining the meaning of the stars of the Newcastle-Ottawa scale in this context

Reply: We have added descriptive headings to all supplemental tables.

Reviewer 2 Report

This work done by Ramai et al. systematically reviewed 19 studies, including 7,381 patients with porphyria, to conclude that patients with porphyria are at risk of developing HCC, with an incidence rate of nearly 5%.

Overall, the idea of the work is interesting, the methodology was run correctly, and the results/recommendations can have merit in the related field. Just a few concerns should be addressed.

 Abstract

Lines 30 and 31: “Interestingly, progression to HCC was seen despite the development of advanced liver disease or cirrhosis” needs to be revised

Methods

Lines 111/112 and 116: please revise the duplication of the work by different coauthors “two authors (DR, AF) did the quality scoring independently” versus “Two reviewers (SD and SC) independently carried out the quality scoring.”

Results

- Figure 1: the number of studies excluded was 31, NOT 29.

- Suppl Table S3: the authors should clarify what is meant by the Asterix (*) in the table footer, and it is better to add an extra column to the left to calculate the total score for each included study.

- A paragraph, including a summary of the quality check results of the enrolled studies, is recommended to be added to the results section.

Limitations

Did the authors find non-English articles during their search? If so, they should include this issue as one of the work limitations (inclusion of only English articles).

Author Response

Reviewer 2:

Comments and Suggestions for Authors

This work done by Ramai et al. systematically reviewed 19 studies, including 7,381 patients with porphyria, to conclude that patients with porphyria are at risk of developing HCC, with an incidence rate of nearly 5%.

Overall, the idea of the work is interesting, the methodology was run correctly, and the results/recommendations can have merit in the related field. Just a few concerns should be addressed.

 Abstract

Lines 30 and 31: “Interestingly, progression to HCC was seen despite the development of advanced liver disease or cirrhosis” needs to be revised

Reply: Thank you, we have made this sentence clearer.

Methods

Lines 111/112 and 116: please revise the duplication of the work by different coauthors “two authors (DR, AF) did the quality scoring independently” versus “Two reviewers (SD and SC) independently carried out the quality scoring.”

Reply: We have amended this duplication.

Results

- Figure 1: the number of studies excluded was 31, NOT 29.

Reply: We have corrected this error.

- Suppl Table S3: the authors should clarify what is meant by the Asterix (*) in the table footer, and it is better to add an extra column to the left to calculate the total score for each included study.

- A paragraph, including a summary of the quality check results of the enrolled studies, is recommended to be added to the results section.

Reply: We have clarified these meanings and amended the supplementary table. We have also added quality data to the results section.

Limitations

Did the authors find non-English articles during their search? If so, they should include this issue as one of the work limitations (inclusion of only English articles).

Reply: For our study, we only included English articles. We have added a line about the generalization of these results to non-English patients.

Reviewer 3 Report

This manuscript is a very careful review of the risk of HCC in patients with porphyria.
I don't think there are any major mistakes or major corrections.

But why does this manuscript not include studies in Asia, where the incidence of HCC is high?
Is it possible to include studies in Asia and other regions in this manuscript, except because it was discarded for screening?
Or is it possible to clearly state in the title or method that the focus was on Europe and the United States from the beginning, or to state as a limitation the reason for not including areas other than Europe and the United States?

Author Response

Reviewer 3:

Comments and Suggestions for Authors

This manuscript is a very careful review of the risk of HCC in patients with porphyria.
I don't think there are any major mistakes or major corrections.

But why does this manuscript not include studies in Asia, where the incidence of HCC is high? Is it possible to include studies in Asia and other regions in this manuscript, except because it was discarded for screening?
Or is it possible to clearly state in the title or method that the focus was on Europe and the United States from the beginning, or to state as a limitation the reason for not including areas other than Europe and the United States?

Reply: Thank you for this very important question. Our study criteria only included articles written/published in English. We did not exclude an article published in Asia or any other region, as long as the article was written in English. To this end, even with the help of a librarian, our search did not yield such articles. We have added a comment on this within the limitations section of the manuscript.

Round 2

Reviewer 1 Report

Thanks for addressing my points. I think the manuscript has improved and is now ready for publication.

Author Response

Many thanks for your review of this paper.